# A Soluble Platelet-Derived Growth Factor Receptor-β Originates via Pre-mRNA Splicing in the Healthy Brain and Is Upregulated during Hypoxia and Aging

**DOI:** 10.3390/biom13040711

**Published:** 2023-04-21

**Authors:** Laura Beth Payne, Hanaa Abdelazim, Maruf Hoque, Audra Barnes, Zuzana Mironovova, Caroline E. Willi, Jordan Darden, Clifton Houk, Meghan W. Sedovy, Scott R. Johnstone, John C. Chappell

**Affiliations:** 1Fralin Biomedical Research Institute (FBRI) at Virginia Tech-Carilion (VTC), Roanoke, VA 24016, USA; 2FBRI Center for Vascular and Heart Research, Roanoke, VA 24016, USA; 3Department of Biomedical Engineering and Mechanics, School of Biomedical Engineering and Sciences, Virginia Tech, Blacksburg, VA 24061, USA; 4Virginia Tech Carilion School of Medicine, Roanoke, VA 24016, USA; 5Department of Biological Sciences, Virginia Tech, Blacksburg, VA 24061, USA

**Keywords:** platelet-derived growth factor receptor-β, pericytes, brain, hypoxia, aging, pre-mRNA splicing

## Abstract

The platelet-derived growth factor-BB (PDGF-BB) pathway provides critical regulation of cerebrovascular pericytes, orchestrating their investment and retention within the brain microcirculation. Dysregulated PDGF Receptor-beta (PDGFRβ) signaling can lead to pericyte defects that compromise blood-brain barrier (BBB) integrity and cerebral perfusion, impairing neuronal activity and viability, which fuels cognitive and memory deficits. Receptor tyrosine kinases such as PDGF-BB and vascular endothelial growth factor-A (VEGF-A) are often modulated by soluble isoforms of cognate receptors that establish signaling activity within a physiological range. Soluble PDGFRβ (sPDGFRβ) isoforms have been reported to form by enzymatic cleavage from cerebrovascular mural cells, and pericytes in particular, largely under pathological conditions. However, pre-mRNA alternative splicing has not been widely explored as a possible mechanism for generating sPDGFRβ variants, and specifically during tissue homeostasis. Here, we found sPDGFRβ protein in the murine brain and other tissues under normal, physiological conditions. Utilizing brain samples for follow-on analysis, we identified mRNA sequences corresponding to sPDGFRβ isoforms, which facilitated construction of predicted protein structures and related amino acid sequences. Human cell lines yielded comparable sequences and protein model predictions. Retention of ligand binding capacity was confirmed for sPDGFRβ by co-immunoprecipitation. Visualizing fluorescently labeled sPDGFRβ transcripts revealed a spatial distribution corresponding to murine brain pericytes alongside cerebrovascular endothelium. Soluble PDGFRβ protein was detected throughout the brain parenchyma in distinct regions, such as along the lateral ventricles, with signals also found more broadly adjacent to cerebral microvessels consistent with pericyte labeling. To better understand how sPDGFRβ variants might be regulated, we found elevated transcript and protein levels in the murine brain with age, and acute hypoxia increased sPDGFRβ variant transcripts in a cell-based model of intact vessels. Our findings indicate that soluble isoforms of PDGFRβ likely arise from pre-mRNA alternative splicing, in addition to enzymatic cleavage mechanisms, and these variants exist under normal physiological conditions. Follow-on studies will be needed to establish potential roles for sPDGFRβ in regulating PDGF-BB signaling to maintain pericyte quiescence, BBB integrity, and cerebral perfusion—critical processes underlying neuronal health and function, and in turn, memory and cognition.

## 1. Introduction

The cerebrovasculature is essential for sustaining all of the unique cell types within the brain [1]. Neurons in particular depend on adequate perfusion to replenish oxygen and nutrients during synaptic communication, as well as to regulate cerebrospinal fluid composition [2]. Therefore, disruption of brain vasculature compromises neuronal health and function, and exacerbates neurodegeneration that fuels cognitive and behavioral deficits, including memory loss and dementia [3,4]. Among the vascular cell types that can become defective within the brain are pericytes [5]. Pericytes are found deep within cerebral capillary networks, extending along the endothelium and encased in the extracellular matrix that forms the vascular basement membrane or basal lamina [6,7]. Recent studies have suggested vessel-associated pericytes contribute to the blood-brain barrier (BBB), where they “tune” vessel permeability to regulate exchange between the circulation and central nervous system (CNS) [8,9]. Pericyte investment within the brain microcirculation depends in part on tight regulation of platelet-derived growth factor-BB (PDGF-BB) signaling through PDGF receptor-beta (PDGFRβ) on the pericyte cell surface [10,11]. Perturbation of the PDGF-BB pathway can undermine pericyte coverage of cerebral microvessels and disrupt vascular function, especially the regulation of BBB permeability and trafficking [12]. These defects can thereby impair neural activity and longevity [13], damaging cognition and memory.

Pericytes are critical cellular constituents of cerebrovascular networks, as discussed above, playing a variety of roles in sustaining brain health and function [14]. Maintaining BBB stability has emerged as a prominent function for cerebral pericytes, as synaptic communication and neuronal health heavily depend on the tight regulation of the brain microenvironment [15]. Vessel leakage and disrupted vesicular transport across the brain endothelium have been attributed in part to pericyte dysfunction and loss [8,16]. Furthermore, pericytes have been implicated in regulating blood flow within brain capillaries, though not without controversy [17,18]. This particular function would significantly impact the oxygen and nutrient delivery to active neurons. Disruption of these and other presumptive pericyte functions would thus compromise brain homeostasis, leading to neuronal dysfunction and degeneration [19]. These injuries in turn fuel the onset and progression of cognitive and behavioral deficits, including memory loss, among other manifestations [20]. It is therefore critical to better understand the molecular determinants that support pericyte function within the cerebral microcirculation, especially regulators within the PDGF-BB pathway [13], given its pleiotropic and potent effects on pericyte biology.

Akin to vascular endothelial growth factor-A (VEGF-A) for endothelial cells, PDGF-BB is considered a “master regulator” of pericyte activity, including proliferation, survival, chemotaxis/migration, and retention within the capillary wall, among other behaviors [21]. Both VEGF-A and PDGF-BB ligands signal via corresponding transmembrane receptor tyrosine kinases (RTKs)—a class of signaling receptors commonly regulated by naturally occurring soluble counterparts [22]. VEGF Receptor-1 (VEGFR1/Flt-1), for example, largely acts as a “decoy” receptor for VEGF-A [23,24]. A soluble isoform of this receptor (sVEGFR1/sFlt-1) generated by pre-mRNA alternative splicing [25] modulates VEGF-A gradients [26], among other functions [27]. Soluble RTKs may also be released from the membrane by proteolytic cleavage [28], which has been described for sVEGFR-1/sFlt-1 [29]. Release of a soluble isoform of PDGFRβ (sPDGFRβ) by proteolytic cleavage has also been recently reported in scenarios of cerebral mural cell hypoxia and damage [5,30,31,32,33]; however, sPDGFRβ production by pre-mRNA alternative splicing remains relatively underexplored.

Soluble isoforms of RTKs can be generated by a variety of mechanisms and in a broad range of neuropathological scenarios. Soluble VEGFR-1/Flt-1, for instance, has recently been implicated in cognitive dysfunction and perturbations in brain structure [34], though the underlying mechanism remains ill-defined. As mentioned above, many RTKs including sVEGFR-1/sFlt-1 can shed extracellular domains via ligand-induced proteolytic cleavage. However, many angiogenic RTKs such as VEGFR1 and VEGFR2/Flk-1 produce high levels of soluble counterparts via pre-mRNA alternative splicing as a mechanism to regulate intracellular signaling [22,26,35,36,37,38]. Both soluble VEGF-A receptors are produced by a skipped splice event, followed by an intronic read-through and in-frame termination. This mechanism frequently occurs with RTKs to create soluble “decoy” receptors that (i) lack a transmembrane domain but retain ligand-binding capacity and (ii) can tether to the extracellular matrix (ECM) through heparin-binding domains [22,39]. These soluble negative regulators can be essential to maintain ligand signaling within a critical physiological range, as seen in the VEGF-A pathway, where loss of sFlt-1 leads to endothelial disorganization, vascular dysmorphogenesis, and embryonic lethality through a variety of defects [23,26,40]. Corresponding mechanisms within the PDGF-BB pathway are still emerging [41] alongside a recent surge in appreciation for PDGF-BB-mediated pericyte dysfunction, particularly within cerebral blood vessels.

Here, we explore one potentially essential regulatory feedback element within the PDGF-BB pathway that likely impacts microvascular pericytes and their function within the brain microvasculature. Soluble PDGFRβ has been identified as a cleavage product from vascular mural cells, and cerebral pericytes in particular [31,32]; yet, to our knowledge, sPDGFRβ production by pre-mRNA alternative splicing has not been widely reported. In the current study, we detected sPDGFRβ protein by Western blot in the brain and in a variety of other murine tissues. We selected the mouse brain for follow-on identification of mRNA sequences that correspond to soluble isoforms of PDGFRβ, generating predicted protein structures and related amino acid sequences. Comparable sequences and protein model predictions were also identified using a human mural cell line. Applying co-immunoprecipitation (co-IP) approaches, we also found that sPDGFRβ retains its capacity for ligand binding, an important characteristic for its potential functional relevance. Custom RNA-Scope^®^ probes allowed for fluorescent in situ hybridization and visualization of sPDGFRβ isoform transcript distribution in mouse brain pericytes adjacent to cerebrovascular ECs. In certain brain regions, we were also able to image sPDGFRβ protein distribution throughout the brain parenchyma and adjacent to certain cerebral microvascular networks, such as near the lateral ventricles. In complementary models, we also found that sPDGFRβ transcript and protein levels increased in mice with age, and that hypoxia can also induce elevated production of sPDGFRβ isoform transcripts and protein.

## 2. Materials and Methods

### 2.1. Animal Care and Use Approval

All experiments involving animal use were performed following review and approval from the Institutional Animal Care and Use Committee (IACUC) at Virginia Tech. All experimental protocols were reviewed and approved by Virginia Tech Veterinary Staff and the IACUC. The Virginia Tech NIH/PHS Animal Welfare Assurance Number is A-3208-01 (Expires: 31 July 2025).

### 2.2. Western Blotting

Wild-type (WT, c57BL/6) mice were euthanized at select time points (P7, n = 3, both sexes; P21, n = 3, both sexes; P90, n = 3, both sexes) by asphyxiation to isoflurane (Cat. No. NDC 13985-030-60, MWI Veterinary Supply Co., Boise, ID, USA) and thoracotomy. Liver incisions allowed for immediate collection of whole blood, and for improved cardiac perfusion with Dulbecco’s phosphate buffer solution (PBS). Brain removal was performed following cardiac perfusion. Specifically, the calvarium was exposed by sharp dissection and incised along the sagittal and lambdoid sutures using angled scissors. The resulting four cranial flaps were reflected away from the brain using forceps. Brains were severed from the spinal cord and cranial nerves using a spatula, and isolated brains were removed. Internal organs (heart, kidney, liver, intestines) were then excised, followed by collection of skeletal muscles from the hindlimb (gracilis and adductor).

Serum was fractionated from whole blood and treated to remove high-abundance proteins such as albumin and immunoglobulins (BioTechne, MIDR002, Minneapolis, MN, USA). All tissues were harvested or transferred into standard Radio Immuno Precipitation Assay (RIPA) buffer and homogenized. Lysates were clarified and quantified using a Bradford Protein Assay (BioRad, Hercules, CA, USA). Samples were prepared for SDS-PAGE in LDS sample buffer (ThermoFisher Scientific, Waltham, MA, USA) and separated on a 4–12% gradient BOLT gel (ThermoFisher Scientific, Waltham, MA, USA). Proteins were transferred to a PVDF membrane using the Trans-Blot Turbo Transfer System (BioRad, Hercules, CA, USA). EveryBlot Blocking Buffer (BioRad, Hercules, CA, USA) was used to block the membrane and dilute antibodies. Primary antibodies were sequentially incubated or together following the application of a stripping buffer. Primary antibody, polyclonal goat anti-mouse PDGFRβ (R&D, AF1042), was used at 1:500 overnight at 4 °C. Primary housekeeping antibodies—mouse anti-mouse α-tubulin (Sigma, T6199-25UL, St. Louis, MO, USA) or rabbit anti-mouse β-actin (Cell Signaling, Danvers, MA, USA, 4967) were used at 1:5000 or 1:500, respectively. Following washes in TBS + 0.05% Tween-20 (TBS-T), secondary antibodies were incubated at room temperature for 1 h, specifically: donkey anti-goat AlexaFluor647 (1:2000, Jackson ImmunoResearch, West Grove, PA, USA, 705-605-147), donkey anti-goat AlexaFluor488 (1:1000, Jackson ImmunoResearch, West Grove, PA, USA, 705-545-003), donkey anti-rabbit AlexaFluor647 (1:1000, Jackson ImmunoResearch, West Grove, PA, USA, 711-605-152), or donkey anti-mouse AlexaFluor647 (1:5000, Abcam, Cambridge, UK, ab150107). Following washes, the membrane was imaged on a Chemidoc^®^ gel imaging system. For sequential target probing, the imaged membrane was stripped using ReBlot Plus Strong Antibody Stripping Solution (Sigma, St. Louis, MO, USA), according to the manufacturer’s instructions. The stripped membrane was re-blotted for a “housekeeping” protein using 1:5000 monoclonal mouse anti-mouse α-tubulin (Sigma, St. Louis, MO, USA, T6199-25UL), overnight at 4 °C, followed by secondary staining with 1:5000 anti-mouse AlexaFluor647 (Abcam, Cambridge, UK, ab150107) and imaging as above. Bands were quantified using volumetric analysis in BioRad ImageLab software (v6.0.1) or integrated density quantification in ImageJ/FIJI.

### 2.3. RNA Ligase-Mediated Rapid Amplification of 3′ and 5′ cDNA Ends (RLM-RACE)

*3′ Ends:* Unique *Pdgfrβ* transcript variants were elucidated using the GeneRacer™ Kit (Invitrogen). A whole brain was harvested into Trizol from a postnatal day 90 (P90) female WT (c57BL/6) mouse (n = 1, female), following a thoracotomy and PBS perfusion as described above. Following homogenization, the tissue was processed for RNA isolation and DNase treatment (Zymo Quick-RNA Miniprep). RNA integrity was confirmed via 28S and 18S levels from agarose gel electrophoresis. Reverse transcription was performed with SuperScript III RT and Oligo dT primers. cDNA ends were amplified via touchdown PCR using Platinum Taq High Fidelity Polymerase (Invitrogen) with the GeneRacer 3′ Primer and a forward (FW) gene-specific primers (GSPs), alongside 3 separate negative controls: no template, no GSP, no GeneRacer 3′ primer. Forward GSPs were designed within Exons 4, 5, and 6:
FW GSP-Exon 4CCCTACGACCACCAGCGAGGTTTCFW GSP-Exon 5CGAGAGCATCACCATCCGGTGCATTGFW GSP-Exon 6TGCCCTCCCGCATTGGCTCCATCCT

Following confirmation via agarose gel electrophoresis, Nested PCR was performed on each PCR amplified product using the GeneRacer Nested 3′ primer and the following FW GSP primers:
FW GSP-Exon 5 (*nested on GSP Exon 4 product*)CGAGAGCATCACCATCCGGTGCATTGFW GSP-Exon 6 (*nested on GSP Exon 5 product*)TGCCCTCCCGCATTGGCTCCATCCTFW GSP-Exon 7 (*nested on GSP Exon 6 product*)GGACGCTGCGGGTGGTGTTCGAGGCTTAT

Products were separated via agarose gel electrophoresis. Individual gel-purified bands (S.N.A.P. columns, Invitrogen) were cloned into the pCR 4-TOPO vector and transformed into One Shot Top10 chemically competent cells and plated overnight. Six colonies were picked from each plate and grown for 16 h in LB broth culture. Plasmids were purified (Monarch Plasmid Miniprep Kit, New England Biolabs, Ipswich, MA, USA), and Sanger sequenced (Virginia Tech Genomics Sequencing Center). Reads were analyzed (DNAstar) and referenced to the mouse *Pdgfrβ* genomic sequence, located within chromosome 18, NC_000084.6. One unique variant was detected, identified in each amplification product, truncating in intron 10 (i10 splice variant). To determine if unique *Pdgfrβ* transcript variants were present in human sources, the described 3′RACE procedure was performed using RNA from human aorta smooth muscle cell lysates. RNA was isolated as described above. Two unique variants were detected, truncating in intron 4 (i4 splice variant) and in intron 10 (i10). The following GSP primers were used:
FW GSP-Exon 3CCTCACTGGGCTAGACACGGGAGAAFW GSP-Exon 4 (*also nested onGSP Exon 3 product*)CTCACTGGGCTAGACACGGGAGAAFW GSP-Exon 6 (*also nested on GSP Exon 4 product*)CCTCACTGGGCTAGACACGGGAGAAFW GSP-Exon 7 (*nested on GSP Exon 6 product*)CCTGGGAGAGGTGGGCACACTACAA

*5′ Ends:* The 5′ end of the mouse *Pdgfrβ* i10 splice variant was elucidated using the GeneRacer™ Kit (Invitrogen, Carlsbad, CA, USA). The P90 WT (c57BL/6) female whole brain RNA (described above) was dephosphorylated and decapped, followed by the protocol described above for 3′ end. To elucidate the 5′ end specific to the i10 variant, the following reverse (Rev) gene-specific primers (GSP) were designed within the unique 3′UTR of the i10 variant, with upstream nested primers:
Rev GSP–i10_aGGCTCAGTACGGCGGGATCAAGGAARev GSP–i10_bTCAGGCTCAGTACGGCGGGATCAARev GSP–Exon 7 (*nested on GSP i10_a product*)CCGGAGTCACCCAAGGTACGGTTGTRev GSP–Exon 6 (*nested on GSP i10_b product*)CGGGAGGGCACTCCAAAGAGGTAGT

Two 5′ ends were detected–a short Exon 1 (truncated upstream) and a unique Exon 1 derived from intron 1 sequence.

### 2.4. Protein Modeling and Functional Domain Mapping

Following elucidation of transcript sequences via RLM-RACE and Sanger sequencing, sequence translations (minus signal sequence) were submitted to the I-TASSER server1–3 for protein structure modeling predictions. The resulting PDB files were submitted to the Cluspro 2.0 server4–7 to predict protein-protein docking with PDGF-BB propeptides (I-TASSER PDB files derived from NM_002608 and NM_011057 for human and mouse, respectively) and heparin binding (selection within Cluspro 2.0). Models were processed using PyMOL2.

The functional PDGFRβ protein domains were mapped to the *Pdgfrβ* exon structure using the UniProt database (for protein) and DNASTAR Lasergene software (for transcripts). Specifically, human domains from accession P09619 were mapped to NM_002609, and mouse domains from P05622 were mapped to NM_001146268. The functional domains tightly corresponded between human and mouse. The crystal structure of complexed human PDGF-BB:PDGFRβ Ig domains 1–3 (D1–D3), PDB-3MJG8, was examined (PyMOL2) to confirm the Uniprot functional IgG 1–3 domain regions, and to decipher ligand binding regions.

### 2.5. Co-Immunoprecipitation (Co-IP) Assay

Co-IP was performed using the Dynabeads Co-Immunoprecipitation Kit (ThermoFisher Scientific, Waltham, MA, USA), using polyclonal goat anti-mouse PDGF-BB antibody (Novus NBP1-52533) for immunoprecipitation. Briefly, a whole brain from a P21 WT (c57BL/6, n = 1) male mouse was harvested and homogenized in immunoprecipitation buffer, following a thoracotomy and PBS perfusion. Dynabeads were coupled with 10 ug of antibody per mg of beads. Whole-brain lysate (1.5 g protein) was processed with 1.5 g coupled beads according to the manufacturer’s instruction, and using the following extraction buffer: 100 mM NaCl, EDTA-free HALT Protease\Phosphatase Inhibitor (ThermoFisher Scientific, Waltham, MA, USA), 2 mM MgCl_2_, and 1 mM DTT. Aliquots were collected at each step for downstream analysis. Eluted protein was immunoblotted under denaturing conditions (see Western blotting protocol above), using 1:1000 polyclonal goat anti-mouse PDGFRβ (R&D, AF1042), and 1:2000 donkey anti-goat AlexaFluor488 (Jackson ImmunoResearch, West Grove, PA, USA, 705-545-147).

### 2.6. Fluorescent RNA-Scope^®^ mRNA Labeling and Confocal Microscopy

Full-length *Pdgfrβ*, i10 *Pdgfrβ* splice variant, and *Pecam1* transcripts were detected in situ using the RNA-Scope^®^ Multiplex Fluorescent v2 Assay (ACD, BioTechne, Minneapolis, MN, USA), according to recommended instructions. Briefly, whole brains were harvested from P21 WT (c57BL/6) mice (n = 4, both sexes), following thoracotomy and PBS perfusion, and were immediately snap frozen in optimal cutting temperature (OCT) compound. Slides were prepared with single 20-micron-thick cryo-slices. Samples were fixed (by 4% paraformaldehyde in PBS) and dehydrated. Following slide preparation, probes were hybridized to target RNA, specifically targeting: (a) i10 *Pdgfrβ* unique sequence in intron 10 (custom designed by ACD, 1148911-C3); (b) full-length *Pdgfrβ* sequence from Exon 13–21, corresponding to intracellular kinase domains (custom designed by ACD, 1148921-C2); and (c) *Pecam1* (pre-designed ACD, 316721). Positive control probes were run for each channel (*Polr2a*-C1, *Ppib*-C2, and *Ubc*-C3), as well as a negative control in each channel, *DapB*, targeting a bacterial gene. Signals were developed and amplified, using 1:1500 Opal Dye 520 for C1-*Pecam1*, 1:1500 Opal Dye 570 for C3-i10_*Pdgfrβ*, and 1:2000 TSA Plus Cy5 for C2-*Pdgfrβ* (Akoya, FP1487001, FP1488001, NEL745001, respectively). High-resolution volumetric images were acquired on a Zeiss LSM 880 confocal microscope. Quantification of RNA-Scope^®^ signal colocalization was performed using ImageJ/FIJI software on single z-plane images, following the application of the “IsoData” threshold filter. Pearson’s colocalization coefficient was obtained using the “Colocalization” plugin in ImageJ/FIJI.

### 2.7. Immunohistochemistry and Confocal Microscopy

Normal WT brains (C57BL/6–P1, n = 4–5, both sexes; P21, n = 3, both sexes) collected for immunohistochemistry and confocal microscopy were collected in a similar manner as described above for protein collection with the exception of intracardiac perfusion of 4% paraformaldehyde (in PBS) following PBS perfusion. Prior to sectioning, the olfactory bulbs and cerebellum were removed from the cerebrum using a single-edged razor blade (Garvey, Camberwell, Australia, 40475). Brains were then caudally affixed by superglue (Loctite, Düsseldorf, Germany, 1364076) to the vibratome dish with their ventral surface supported by a block of 4% agarose. They were then submerged in PBS and sectioned into 100 μm slices using half a double-edged razor blade (Electron Microscopy Sciences, Hatfield, PA, USA, 72000) in a vibratome (1000-Plus, Pelco 102, Ted Pella Inc., Redding, CA, USA). Slices were transferred to a 24-well plate and then stored in PBS at 4 °C until processing for immunohistochemistry.

Brain sections were blocked for 1 h at room temperature in PBS and 0.1% Triton 100x (PBS-T) with 1.5% Normal Donkey Serum (Jackson ImmunoResearch, West Grove, PA, USA, 017-000-121). Slices were then incubated at 4 °C overnight in 1.5% Normal Donkey Serum in PBS-T with rat anti-PDGFR-β antibody (Affymetrix, Santa Clara, CA, USA, 14-1402, 1:500), goat anti-CD31/platelet-endothelial cell adhesion molecule-1 (PECAM-1) antibody (R&D Systems, Minneapolis, MN, USA, AF3628, 1:500), and rabbit anti-glial-fibrillary acidic protein (GFAP) (Millipore Sigma, Burlington, MA, USA, AB5804, 1:500) on a shaker. After incubation with primary antibodies, the slices were washed four times for 15 min at room temperature in PBS-T on a shaker.

Brain slices were then incubated for either 4 h at room temperature or overnight at 4 °C in 1.5% Normal Donkey Serum in PBS-T with donkey anti-rat DyLight 550 (ThermoFisher, Waltham, MA, USA, SA5-10029, 1:1000), donkey anti-goat AlexaFluor488 (Jackson ImmunoResearch, West Grove, PA, USA, 705-545-147, 1:500), donkey anti-rabbit AlexaFluor647 (Jackson ImmunoResearch, West Grove, PA, USA, 711-605-152, 1:1000), and 4′,6-Diamidino-2-phenylindole (DAPI, Sigma, Burlington, MA, USA, D9542, 1:1000). After incubation with secondary antibodies, the slices were washed four times for 15 min at room temperature in PBS-T, and then washed again for 15 min at room temperature in PBS after final incubations of staining combinations. Stained brain slices were mounted in 50% glycerol in PBS, a coverslip applied (22 mm × 22 mm–1.5 thickness, ThermoFisher, Waltham, MA, USA, 12-541-B), and sealed with clear nail polish (Electron Microscopy Sciences, Hatfield, PA, USA, 72180). Images of stained brain sections were acquired with a Zeiss LSM 880 confocal microscope using a 20× objective lens and Zen Black software.

Subcortical regions were identified as being above ventricles and included cortex and subcortical white matter. Images were collected in 2 × 2 tile scans with 10% overlap in 50-100 *z*-axis sections as determined by optimal *z*-axis distance. Using Zen Black image processing capability, the original tile scans were stitched together. Selected brain regions of interest were outlined and saved as new image files for analysis.

### 2.8. Quantitative Real-Time Polymerase Chain Reaction (qRT-PCR)

Following thoracotomy and PBS perfusion, tissues were harvested into Trizol from WT (c57BL/6) mice (equally distributed by sex, when possible) at each of three timepoints: P5 (n = 4, both sexes), P21 (n = 5, both sexes), P90 (3-months-old, n = 4, both sexes). To capture a time-point associated with aging, tissues were collected from 20-month-old mice (n = 4, both sexes) which, while not of the same WT background, was considered as a littermate control for parallel experiments. Specifically, this mouse was negative for Cre-recombinase, but harbored floxed STOP codons within the *Vhl* gene (i.g. *UBC-CreER negative; Vhl^lox/lox^* on the CD1 background) and exhibited no phenotypic changes by various metrics [42]. RNA was isolated using the Zymo Direct-zol RNA Mini-prep kit, with DNase treatment. Isolated RNA was further treated for genomic material using a Turbo DNA-Free kit (ThermoFisher Scientific, Waltham, MA, USA). Samples were reverse transcribed using a High-Capacity cDNA RT Kit (Applied Biosystems, Waltham, MA, USA). Each sample was run in triplicate for qRT-PCR (QuantStudio Flex 6). To quantify full-length *Pdgfrβ* transcripts, Applied Biosystem Taqman assay mix Mm01262481_g1 (Fisher Scientific, Waltham, MA, USA) was used, which targets the exon 13-14 boundary. Custom probes were designed and validated to quantify the *Pdgfrβ* i10 splice variant:
Forward PrimerAACTCCATGGGTGGAGATTCReverse PrimerGTGAGAGTCATCAGAGCCATCProbe SequenceTCACCGTGGTCCCACATTGTGAG

### 2.9. Cell Culture and Hypoxia Exposure

Primary human coronary artery smooth muscle cells (cASMC, #PCS-100-021, ATCC) were cultured in human vascular smooth muscle cell basal media (#M231500, ThermoFisher, Waltham, MA, USA), supplemented with smooth muscle cell growth supplement (SMGS, #S00725, ThermoFisher, Waltham, MA, USA). All cells were used within 16 population doublings, as recommended by the manufacturer. CASMC were grown in six-well plates to 70–80% confluence, washed with PBS twice, and media changed for smooth muscle cell basal media (M231500, ThermoFisher, Waltham, MA, USA) containing 2% FBS for 3 days to induce cell stall. Stalled cells at 100% confluence were treated with 50 ng/mL recombinant human platelet-derived growth factor-BB (PDGF-BB, 50ng/mL, #PHG0045, ThermoFisher, Waltham, MA, USA) for 24 h prior to harvest.

Mouse embryonic stem cells (ESCs) harboring reporters for cells within endothelial and pericyte lineages were generated as previously described [43]. These “double reporter” (DR)-ESCs were maintained as undifferentiated cells through exposure to leukemia inhibitory factor (LIF). Upon LIF removal, DR-ESC differentiation was initiated, and cells were cultured as floating embryoid bodies (EBs) for 3 days. They were then allowed to adhere to tissue-culture-treated plates and differentiate for 7 additional days into primitive vascular structures, along with other cell types. Hypoxia-exposed cultures were incubated in a gas-tight chamber (STEMCELL Technologies) filled with 3% O_2_, 5% CO_2_, and 92% N_2_ for the last 12 or 24 h of their differentiation (n = 4–6). Control DR-ESC cultures were maintained at atmospheric O_2_ (~21%) and 5% CO_2_ for the duration of their differentiation. At the conclusion of the experiment, RNA was collected in Trizol and processed as described above to measure relative levels of full-length and i10 variant *Pdgfrβ* transcripts by qRT-PCR.

### 2.10. Statistical Analysis

Statistical analysis was performed using GraphPad Prism software, version 9.1.2 (226). For the cell-based hypoxia experiment, the data were analyzed by an ordinary one-way ANOVA followed by unpaired Tukey’s *t*-test. Outlier data points were removed by applying the Grubbs’ test, or the extreme studentized deviate (ESD) method, with α = 0.05. Significant differences were determined as P values less than or equal to 0.05.

## 3. Results

### 3.1. Soluble PDGFRβ Is Present across Various Murine Tissues with a Distinct Enrichment in the Blood, Kidney, and Brain

Truncated isoforms of PDGFRβ have been detected across a wide range of biological contexts, from in vitro settings, such as differentiating mouse embryonic stem cells [44], to in vivo scenarios, such as the healthy and diseased mouse and human brain [32,33]. To better characterize the abundance of sPDGFRβ across a variety of tissue compartments, we collected protein from normal, healthy WT (c57BL/6) postnatal day 21 (P21) and adult (P90) murine brain, heart, kidney, intestine, liver, skeletal muscle, and blood serum (Immunoglobulin G depleted). Proteins from each of these organ systems, along with recombinant full-length PDGFRβ protein as a positive control (data not shown), were processed for Western blot and immunolabeling for PDGFRβ (Figure 1). We observed bands corresponding to larger protein weights (100–160 kDa) for the recombinant protein control and for most tissues corresponding to full-length PDGFRβ, with varying levels for each tissue. As expected, serum proteins did not appear to contain much, if any, full-length PDGFRβ at higher molecular weights. In contrast, the serum lanes for both time-points displayed prominent bands at a lower molecular weight corresponding to approximately 32 kDa (Figure 1). Notably, ~32 kDa bands were also found with kidney and brain protein lysates, while the other tissues appeared to contain relatively lower amounts of the truncated PDGFRβ isoform within this size range.

### 3.2. PDGFRβ Is Alternatively Spliced to Generate a Soluble Isoform That Retains PDGF-BB Ligand Binding Capacity

While proteolytic cleavage has been proposed as the primary mechanism for generating soluble PDGFRβ isoforms [31,32], we hypothesized that alternative splicing on the mRNA level might also yield sPDGFRβ isoforms. To test this hypothesis, we collected and isolated high-quality mRNA from a postnatal day 90 (P90) murine brain. We used 3′ rapid amplification of cDNA ends (RACE) to elucidate and Sanger sequence unique 3′ ends of PDGFRβ mRNAs (see Methods for full details). Sequence reads were referenced to the mouse *Pdgfrβ* genomic sequence. One unique variant was detected, identified in each amplification product, truncating in intron 10 (Figure 2, *see Appendix A for nucleotide sequence information*). To determine if a similar *Pdgfrβ* variant exists in human cells, the same RNA ligase-mediated rapid amplification of 3′ cDNA ends (RLM-RACE) was applied to RNA from human smooth muscle cell lysates. From this analysis, two unique variants were detected, truncating in intron 10, as in the mouse, but also in intron 4 (Figure 2). Follow-on work is underway to determine if truncation in intron 4 also occurs in the mouse. We also elucidated the 5′ end of the murine intron 10 (i10) *Pdgfrβ* splice variant to generate a complete sequence for this isoform, facilitating protein modeling and development of additional tools and approaches to detect this variant.

Drawing from the sequence information established for these variants, we sought to generate protein structure modeling predictions to map functional domains. These models allowed us to (i) predict important domains such as PDGF-BB and heparin binding regions, and (ii) compare these domains across species. Sequence translations (without signal sequence) from both putative mouse and human sequences were submitted for protein model creation (Figure 3, *see Appendix A for detailed amino acid predictions*). Functional PDGFRβ protein domains were also mapped to the *Pdgfrβ* exon structure. Functional domains for the murine and human sequences displayed a high degree of homology. Furthermore, protein docking predictions and analysis of known crystal structures of complexed human PDGF-BB with PDGFRβ supported the idea that these isoforms generated by pre-mRNA alternative splicing likely retain their ligand binding capacity. To directly address the potential of sPDGFRβ isoforms to bind PDGF-BB, we performed a co-immunoprecipitation (co-IP) assay. Specifically, we selected a PDGF-BB antibody for protein capture by adsorbing these antibodies to beads, and incubating antibody-coated beads with whole brain protein lysate from a normal, healthy P21 WT (c57BL/6) mouse. Proteins captured by the anti-PDGF-BB antibodies were eluted and immunoblotted under denaturing conditions using PDGFRβ antibody labeling and detection (Figure 3). We detected truncated PDGFRβ variants in the whole brain lysate and in the eluted fraction following co-IP, with minimal to no detection in the depleted flow-through fraction of supernatant following immunoprecipitation. These data are consistent with the notion that soluble isoforms of PDGFRβ retain their capacity to bind PDGF-BB ligands, including sPDGFRβ proteins generated by pre-mRNA alternative splicing.

### 3.3. Pdgfrβ Splice Variant Transcripts Localize near Pecam1-Positive Cells in the Murine Brain with PDGFRβ Protein Associated with Vessels and Interstitial ECM

In addition to facilitating construction of predicted protein models, sequence identification for the murine i10 *Pdgfrβ* splice variant enabled the design and synthesis of custom RNA-Scope^®^ probes to label mRNA transcripts in situ. We hypothesized that *sPdgfrβ* transcripts in the murine brain would spatially correlate with transcripts of full-length *Pdgfrβ* and with cerebrovascular endothelial cells positive for platelet-endothelial cell adhesion molecule-1 (PECAM-1). To test this hypothesis, we collected normal, healthy WT (c57BL/6) P21 murine brains for snap freezing and coronal cryosectioning. Brain cryo-slices were fixed, dehydrated, and prepared for the hybridization of RNA probes against (i) the i10 *sPdgfrβ* variant sequence (see Figure 2 for regions targeted by custom probes), (ii) full-length *Pdgfrβ* sequence corresponding to the intracellular kinase domain, and (iii) *Pecam1*. Positive and negative control probes were applied to parallel samples. Probes were labeled by corresponding dyes with signal amplification, and imaged by high resolution confocal microscopy. Tiled images were collected to spatially assess *sPdgfrβ* transcript distribution relative to full-length *Pdgfrβ* and *Pecam-1* transcripts (Figure 4).

The vast majority of fluorescent signals associated with *sPdgfrβ* transcripts overlapped with full-length *Pdgfrβ* transcript signals, although not entirely, as distinct *sPdgfrβ*-associated signals were noted. These overlapping signals demonstrated a strong localization with the *Pecam1*-positive signals throughout the brain as well, consistent with the spatial configuration of endothelial cells and pericytes within the brain microvasculature (Figure 4), though overlap could represent a subset of double-positive vascular cells derived from hematopoietic lineages [45]. The distribution of these signals also appeared to be fairly conserved across different brain regions, including the cortex, the hypothalamus, and along the floor of the third ventricle, adjacent to the thalamus (e.g., neighboring the paraventricular nucleus of the thalamus) (Figure 4). In addition, we did not observe any notable differences in the spatial distribution of these signals across sexes.

Having mapped the spatial distribution of *sPdgfrβ* transcripts relative to full length *Pdgfrβ* mRNA and *Pecam1*-expressing cells within murine brain sections, we then wanted to assess the distribution and localization of PDGFRβ protein. We have previously immunostained early postnatal murine brains for PDGFRβ (P1 [46], and P5 [47]) to observe pericytes and their interactions with the developing cerebrovasculature. In addition to observing distinct signals on the cell surface of pericytes within these brain slices, we noted regions of diffuse signals with varying intensity associated with the PDGFRβ staining. Here, we replicated those findings by immunolabeling PDGFRβ protein alongside visualization of endothelial cells (anti-PECAM-1 antibody labeling) and glial fibrillary acidic protein (GFAP)-positive astrocytes (Figure 5). In brain sections from normal, healthy P1 and P21 WT (c57BL6) mice, we found PDGFRβ-positive pericytes along brain microvessels located in several areas neighboring the caudothalamic groove in the floor of the lateral ventricle, posterior to the interventricular foramen. However, we also observed regions of PDGFRβ signal located beyond vessel walls that were not associated with PECAM-1-positive endothelial cells, presumably distributed in the surrounding extracellular matrix (ECM) (Figure 5). Additionally, we observed increased cellular density corresponding with ECM-associated PDGFRβ signal. Though we could not exclude the possibility of direct overlap between the PDGFRβ and GFAP signals, obvious convergence of these signals appeared relatively infrequent, suggesting minimal association with astrocytes (Figure 5), which have been proposed to express full-length *Pdgfrβ* under certain conditions [48]. Overall, these observations were consistent with non-cell associated PDGFRβ that likely represents a soluble variant that can tether to the ECM.


*3.4 Aging and Acute Hypoxia Induce an Increase in Transcript and Protein Levels of sPDGFRβ*


Cognitive and behavioral abnormalities such as memory loss and dementia arise in part from neurodegeneration occurring in the context of aging and during hypoxic stress [49]. These factors likely contribute to neuronal damage through, among others, cerebrovascular dysfunction, which is presumably driven by aberrant regulation of cells within the blood vessel wall. Because pericyte regulation by the PDGF-BB signaling axis might be involved in these vascular defects, we sought to address the question of how soluble PDGFRβ variants might be affected by these factors, namely aging and hypoxia. To do this, we collected whole murine brains at P5, P21, P90 (3 months old), and 20 months old. Following RNA isolation and cDNA synthesis, we conducted quantitative RT-PCR (Figure 6) using probes against full-length *Pdgfrβ* transcripts (targeting the exon 13–14 boundary) and the soluble i10 *Pdgfrβ* variant (via custom-designed probes—see Figure 2 for regions targeted by custom probes). We found that full-length *Pdgfrβ* transcripts were more abundant than *sPdgfrβ* transcripts at earlier ages (P5 and P21) (Figure 6); however, by P90, *sPdgfrβ* transcript abundance increased by nearly five-fold relative to the P5 time-point, while full-length *Pdgfrβ* mRNA stayed relatively constant. At 20 months of age, full-length *Pdgfrβ* expression appeared to diminish relative to P5 levels, while *sPdgfrβ* transcript levels continued to increase to nearly 10-11-fold higher relative quantitation than at P5 (Figure 6). This trend was reflected in the relative protein levels of each PDGFRβ isoform at each time point as well. Protein lysates from whole P7, P21, and P90 murine brains were analyzed by Western blot with anti-PDGFRβ and anti-α-tubulin antibodies used for detection. For P7 brains, a distinct band at ~180 kDa was detected, corresponding to full-length PDGFRβ; in contrast, faint signals around the 32 kDa molecular weight were observed, suggesting a relatively low abundance of sPDGFRβ isoforms at this early age (Figure 6). Bands appeared more prominently in this size range for P21 murine brain lysates relative to P7, and signal associated with full-length PDGFRβ also appeared more intense at this time-point. Signal strength for bands in the 32 kDa range intensified further for P90 brain lysates, indicating a potential increase in sPDGFRβ abundance. Bands at the higher molecular weight (~180 kDa) appeared to decrease in density at the P90 age, mirroring the mRNA relative quantitation data (Figure 6).

To test our hypothesis that sPDGFRβ variant levels are oxygen-sensitive, we took advantage of a recently developed cell-based model containing reporter-labeled endothelial cell and pericyte lineages that would permit a high degree of experimental control over the oxygen environment. Specifically, mouse embryonic stem cells (ESCs) were differentiated into endothelial cell and pericyte precursors that give rise to primitive vascular structures composed of these cell types [43]. Following visual confirmation of early vessel formation, we exposed these cultures to hypoxic conditions for 12 and 24 h, and then collected mRNA for quantification of the soluble i10 *Pdgfrβ* variant and full-length *Pdgfrβ*. After 12 and 24 h of exposure to 3% O_2_, soluble *Pdgfrβ* isoforms displayed a nearly three-fold increase in transcript levels relative to the control (i.e., cultures exposed to ambient O_2_ levels until collection) (Figure 6). In contrast, the expression of full-length *Pdgfrβ* remained relatively constant, with a slight decrease at the 24-h time point. Protein levels of a ~32 kDa isoform that stained positive for PDGFRβ remained fairly constant at 12 h of hypoxia (relative to controls), but increased after 24 h of hypoxic conditions (Figure 6). These data suggest that *sPdgfrβ* levels are likely sensitive to variations in oxygen levels, with acute hypoxia inducing a marked increase.

## 4. Discussion

The PDGF-BB pathway is a critical regulator of microvascular pericytes, facilitating their investment and retention within the cerebral microcirculation. Similar RTKs such as VEGF-A are modulated by soluble isoforms of cognate receptors that maintain signaling within a physiological range. Soluble variants of PDGFRβ have been reported, with their origins being attributed to enzymatic cleavage from the surface of vascular mural cells, and specifically cerebral pericytes [31,32]. Here, we present evidence that sPDGFRβ is also generated by pre-mRNA alternative splicing, terminating prior to the transmembrane and intracellular domains. We detected truncated PDGFRβ isoforms in a range of murine tissues, including the blood (serum), kidney, and brain. Given the importance of PDGFRβ signaling for cerebral pericytes, we utilized brain tissue for follow-on analysis, including sequence identification and protein model predictions. Corresponding sequences and protein models were also found in human cells, suggesting a conservation across species. Co-immunoprecipitation of truncated/soluble PDGFRβ with PDGF-BB corroborated protein-protein docking model predictions and crystal structure data (human PDGF-BB with PDGFRβ domains 1–3), supporting the likely essential functional relevance of sPDGFRβ. In situ hybridization and fluorescent labeling of *sPdgfrβ* transcripts in murine brain revealed a strong, though not complete, colocalization with full-length *Pdgfrβ* transcripts. Soluble *Pdgfrβ* transcripts were also in close proximity to transcripts labeled for the endothelial cell marker *Pecam1*, consistent with cerebrovascular mural cells as a source of sPDGFRβ under normal physiological conditions. Visualization of immunofluorescently labeled PDGFRβ protein distribution throughout certain brain regions suggested the presence of ECM-associated PDGFRβ protein in the brain parenchyma, in addition to vessel-associated pericytes. We also took initial steps to understand factors that might regulate the production of *sPdgfrβ* variants, and found that aging and hypoxia were associated with an increased abundance of i10 soluble PDGFRβ isoforms relative to full-length *Pdgfrβ* expression. Overall, our data are consistent with the idea that (i) soluble PDGFRβ isoforms are present in the normal murine brain, likely generated by pre-mRNA alternative splicing, and (ii) corresponding *sPdgfrβ* transcripts, which were similarly detected in human cells, are sensitive to aging and hypoxia–two factors that can fuel cerebrovascular dysfunction, neurodegeneration, and in turn, memory loss and dementia.

Truncated isoforms of PDGFRβ have been previously reported, with a range of molecular weights reported for different physiological and pathological contexts [5,30,31,32,33,44]. For instance, in the setting of amyloid-beta exposure or severe hypoxia (1% O_2_), human pericytes are proposed to shed a soluble isoform of PDGFRβ in the size range of ~160 kDa [5], suggested to be an indicator of A Disintegrin And Metalloproteinase domain-containing protein 10 (ADAM10) activity, BBB breakdown, and cognitive decline [32]. Cultured primary mouse pericytes from normal, healthy postnatal (P5) and adult (6 months) brains, however, have also been reported to generate 110 and 160 kDa species of PDGFRβ, while cortical slices from mouse brain generated a 60 kDa PDGFRβ isoform after 2 weeks in organotypic culture [33]. Considering current sequence data from both humans and mice, it appears that the entire extracellular portion of PDGFRβ–post signal cleavage–is around 55 kDa without glycosylation. Therefore, truncated forms of PDGFRβ larger than ~60 kDa (allowing for glycosylation) likely extend through the transmembrane domain (<3 kDa) and into the intracellular region. As we learn more about truncated PDGFRβ isoforms, we will also need to expand our understanding of the potential mechanisms that may give rise to these unique growth factor receptor species. In the current study, we repeatedly detected an ~32 kDa PDGFRβ variant from normal, healthy murine brains, with double bands potentially arising from variable glycosylation. Our sequence analysis found a correspondence between this protein and a *Pdgfrβ* splice variant generated by truncation in intron 4 (i4 splice variant–identified in human cells, and a predicted sequence in mice cells still being resolved). The i10 *Pdgfrβ* splice variant likely corresponds to larger protein isoforms (~50–60 kDa) previously detected by our group [44] and others [33]. We have demonstrated here that sPDGFRβ i10 is expressed during normal physiological conditions, but is altered during hypoxia and aging in the brain. Thus, these findings provide a new perspective for future investigations into PDGFRβ isoforms that are essential for tissue homeostasis versus those associated with pathology, as well as for studies identifying their context-dependent regulatory mechanisms.

The spatial distribution of soluble growth factor receptors often depends on the presence of unique motifs that facilitate binding to the ECM. Soluble Flt-1/VEGFR1, for instance, includes a heparin-binding domain in the fourth Ig-like loop of its protein structure [50], which can be targeted for sFlt-1/sVEGFR1 release from the ECM by exposure to unfractionated heparin in tissues such as the placenta [51]. Sequence analysis in the current study suggests that the previously unreported i10 PDGFRβ splice variant (but not the i4 isoform) retains a heparin binding domain capable of facilitating physical association with the ECM via heparin sulfate interactions. This property is consistent with our observation of a diffuse signal for PDGFRβ immunostaining in murine brain slices in addition to the expected labeling of full-length PDGFRβ on pericyte cell walls. We noted an enrichment of this more dispersed signal in periventricular brain regions. These areas displayed a relatively higher density of cells, as seen from the corresponding nuclear labeling, suggesting an increased concentration of ECM proteins as well. PDGFRβ that may be present on the astrocyte cell wall did not appear to substantially contribute to this diffuse signal, though we could not fully exclude contributions from astrocytes or all possible cell types that may be expressing PDGFRβ in these brain regions. Another phenomenon potentially contributing to elevated sPDGFRβ in these sub-regions may be an increased exposure to PDGF-BB ligands. Soluble receptors associated with RTKs are often generated as a feedback mechanism to maintain growth factor signaling within an appropriate range [22,26,35,36,37,38]. Therefore, their levels frequently increase when cells are exposed to high levels of ligand, and this response may be partially responsible for the noted increase in sPDGFRβ immunostaining signal in periventricular areas. Previous studies have suggested that the production of specific PDGFRβ isoforms may indeed be ligand-sensitive [33], but testing this response for the isoforms identified herein was beyond the scope of the current study. Additional insight into how PDGFRβ isoform levels and distribution may be differentially regulated throughout the brain will be critical for understanding not only their upstream regulation, but also their functional relevance for cerebrovascular homeostasis.

Growth factor receptor isoforms can perform a number of unique functions, depending on the signaling pathway and the context. The soluble epidermal growth factor receptor (sEGFR), for example, can inhibit intracellular signaling in target cells by (i) acting as a “decoy” receptor to prevent ligand binding of surface receptors, or (ii) binding the extracellular domain of full-length EGFRs. Soluble variant interaction with transmembrane receptors can create inactive heterodimers or prevent internalization [52], thereby interfering with initial signaling events. Previous studies have utilized full-length sequence information to artificially create truncated, soluble versions of PDGFRβ [41,53]. These dominant-negative constructs have been introduced into a variety of settings, but their primary effect appears to be antagonizing signaling via full-length PDGFRβ and accordingly altering cell activation and behavior. Truncated PDGFRβ isoforms generated by proteolytic cleavage have been primarily considered biomarkers of pericyte injury or death [31,32], with potential functional roles remaining relatively unexplored. Soluble PDGFRβ variants produced by alternative splicing also remain to be characterized with regard to their functional relevance, but, supported by ligand-binding capacity demonstrated herein, it is likely that they play similar roles as those within other RTK pathways, such as sFlt-1/VEGFR1 [22,26,35,36,37,38]. In pathological scenarios, dysregulated sPDGFRβ interference with signaling via the full-length receptor, particularly for cerebrovascular pericytes, is likely to lead to deleterious effects akin to those reported for sFlt-1/sVEGFR1 [34]. More specifically, sPDGFRβ-mediated attenuation of pericyte signaling via full-length receptors may undermine pericyte investment within cerebral capillaries, leading to BBB and/or vasomotor defects, neuronal dysfunction and degeneration, and ultimately, memory loss and cognitive decline [13]. Thus, as we continue to learn more about how sPDGFRβ isoforms are created and regulated in the brain and beyond, it will be essential to also establish their contributions to both physiological and pathological processes, to better understand their utility as biomarkers of disease and also as potential drug targets.

## Figures and Tables

**Figure 1 biomolecules-13-00711-f001:**
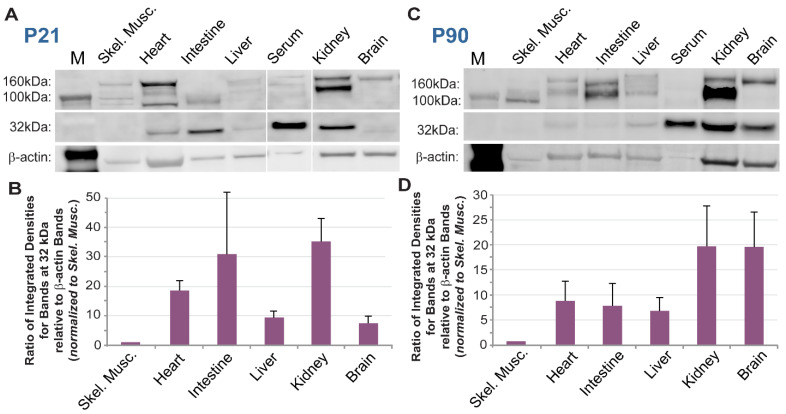
**A Truncated Isoform of PDGFRβ was Detected alongside Full-Length Receptors across a Range of Murine Tissues at Different Ages.** (**A**) Representative images of a Western blot for proteins from select murine tissues at postnatal day 21 (P21) (from n = 3 biological replicates, both sexes) detected by immunolabeling for PDGFRβ and β-actin–lane 1: molecular size marker (M), lane 2: skeletal muscle (Skel. Musc.), lane 3: heart, lane 4: intestine, lane 5: liver, lane 6: blood serum, lane 7: kidney, lane 8: brain. (**B**) Graph of the ratio of integrated densities for bands at the 32 kDa size range relative to β-actin bands for each tissue indicated, all of which were normalized to the skeletal muscle samples. Bars are averages with standard deviations. (**C**) Representative images of a Western blot for proteins from select murine tissues at postnatal day 90 (P90) (from n = 3 biological replicate, both sexes) detected by immunolabeling for PDGFRβ and β-actin–lane 1: molecular size marker (M), lane 2: skeletal muscle (Skel. Musc.), lane 3: heart, lane 4: intestine, lane 5: liver, lane 6: blood serum, lane 7: kidney, lane 8: brain. (**D**) Graph of the ratio of integrated densities for bands at the 32 kDa size range relative to β-actin bands for each tissue indicated, all of which were normalized to the skeletal muscle samples. Bars are averages with standard deviations.

**Figure 2 biomolecules-13-00711-f002:**
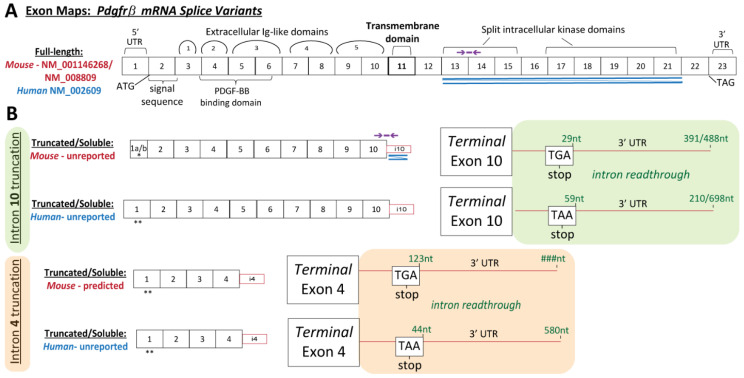
**Exon Maps of Full-Length and Truncated *Pdgfrβ* mRNA Transcripts were Constructed Based on Known and RLM-RACE-Generated Sequences.** (**A**) Full-length *Pdgfrβ* mRNA exon map based on murine (red accession number) and human (blue accession number) sequences with various regions labeled: signal sequence, PDGF-BB binding domain, five extracellular Ig-like domains, the transmembrane domain, and split intracellular kinase domains. Double blue lines indicate the region used to design RNA-Scope^®^ probes for in situ transcript detection. Purple line and arrows indicate the approximate region corresponding to qRT-PCR probes for relative quantitation of mouse full-length *Pdgfrβ* transcripts. (**B**) *Pdgfrβ* mRNA exon maps for soluble isoforms truncated at intron 10 (green background) or intron 4 (orange background) from both mouse and human. Double blue lines indicate the region used to design RNA-Scope^®^ probes for in situ transcript detection; 3′ UTR intron “read-through regions are shown corresponding to each truncation location and species. Purple line and arrows indicate the region used to design custom qRT-PCR probes for relative quantitation of i10 mouse *sPdgfrβ* transcripts. Single asterisk (*) indicates an established 5′ sequence, while double asterisk (**) indicates 5′ sequences that are currently unresolved for these variants.

**Figure 3 biomolecules-13-00711-f003:**
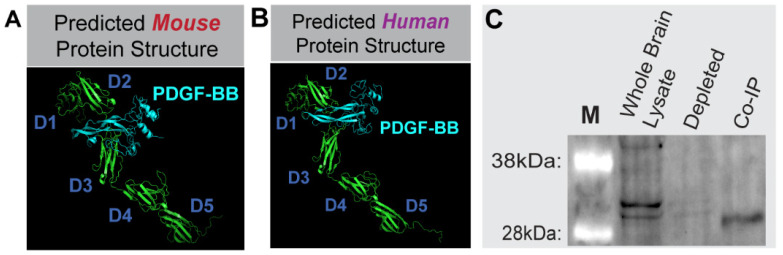
**Soluble PDGFRβ Engagement with PDGF-BB Ligand was Supported by Predicted Protein Models of Mouse and Human Splice Variants and sPDGFRβ Co-Immunoprecipitation using PDGF-BB Capture.** Protein models of mouse (**A**) and human (**B**) sPDGFRβ variants including predictions for PDGF-BB (cyan) binding. “D” denotes major folding domains. (**C**) Immunoblot results (n = 1, P21 male) from the sPDGFRβ co-IP, with “M” denoting size marker. Whole Lysate shown in Lane 2, the depleted fraction in Lane 3, and the co-IP elution labeled for PDGFRβ in Lane 4. Note the distinct bands at ~32 kDa in the Whole Lysate and co-IP lanes.

**Figure 4 biomolecules-13-00711-f004:**
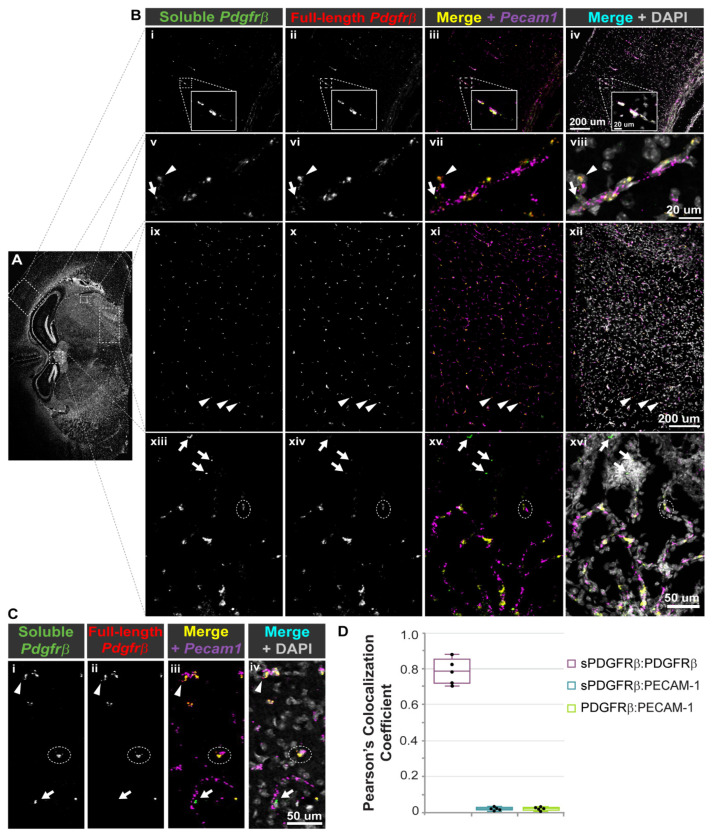
**RNA-Scope^®^ Probes Facilitated Detection of Soluble and Full-length *Pdgfrβ* Transcripts Adjacent to *Pecam1* mRNA in the Adult Mouse Brain.** (**A**) Representative coronal section of an adult mouse brain (from n = 4 biological replicates, both sexes) labeled for cell nuclei (DAPI) with white, dotted boxes indicating regions visualized at a higher-power magnification shown in (**B**). (**B**) High-power magnification images of mouse cerebral cortex and white matter (i–iv with associated inset), thalamus (v–viii), hypothalamus (ix–xii), and along the floor of the third ventricle, adjacent to the thalamus (xiii–xvi) labeled for soluble *Pdgfrβ* transcripts (i, v, ix, xiii; green in iv, viii, xii, xvi), and full-length *Pdgfrβ* transcripts (ii, vi, x, xiv; red in iv, viii, xii, xvi). A merge of these two signals appears yellow in iii, vii, xi, and xv. While most of these signals overlap, we found instances where the signal was stronger for full-length *Pdgfrβ* (arrowheads, ix–xii) or soluble *Pdgfrβ* (arrows, xiii–xvi). *Pecam1* transcripts were also detected (magenta in iii, iv, vii, and viii), along with cell nuclei (DAPI, white in iv and viii), with many locations where *Pecam1*-positive signals spatially correlated with *Pdgfrβ* transcript signals (see dashed white oval, xiii–xvi). Scale bars are 200 microns in iv and xii, 20 microns in the inset in iii and in viii, and 50 microns in xvi. (**C**) Representative high-power images of a mouse brain section labeled with RNA-Scope^®^ probes against soluble *Pdgfrβ* (i, and green in iii and iv), full-length *Pdgfrβ* (ii, and red in iii and iv), and *Pecam1* (magenta in iii and iv), along with cell nuclei (DAPI, white in iv). Most signals converged, though we noted instances of more prominent full-length *Pdgfrβ* (arrowhead) or soluble *Pdgfrβ* (arrow), with signal also corresponding to *Pecam1*-positive signals (dashed white oval). Scale bar is 50 microns. (**D**) Graph of Pearson’s Colocalization Coefficient for each of the signal combinations indicated, specifically sPDGFRβ and full-length PDGFRβ (magenta box, sPDGFRβ:PDGFRβ), sPDGFRb and PECAM-1 (cyan box, sPDGFRb: PECAM-1), and full-length PDGFRb and PECAM-1 (green box, PDGFRb:PECAM-1).

**Figure 5 biomolecules-13-00711-f005:**
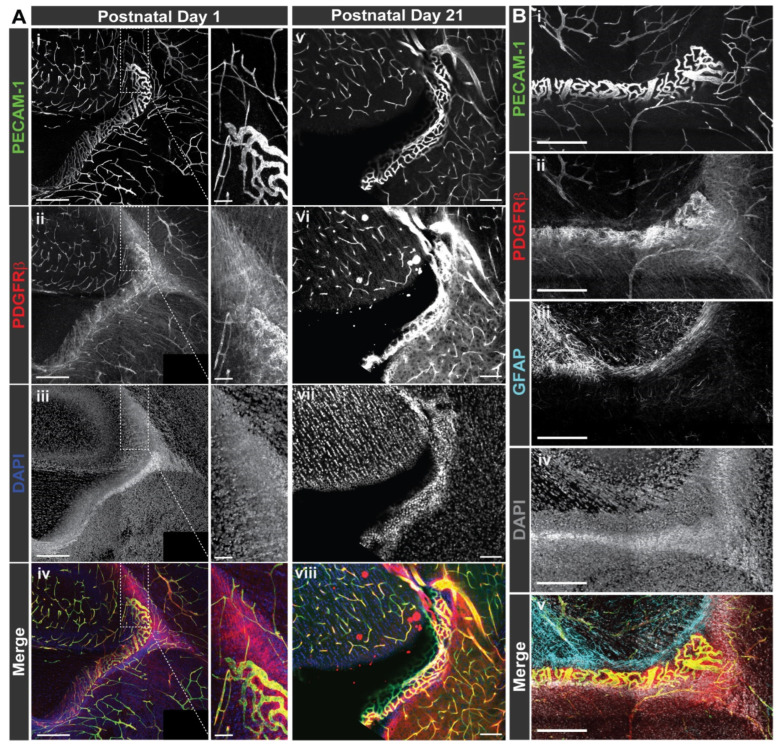
**Immunofluorescent Labeling of PDGFRβ in the Mouse Brain Revealed Signals Alongside PECAM-1-Positive Vessels but also Diffusely Distributed throughout the Parenchyma.** (**A**) Representative confocal images of mouse brain sections immunohistochemically labeled for PECAM-1 (i, v; green in iv, viii) and PDGFRβ (ii, vi; red in iv, viii) with cell nuclei stained with DAPI (iii, vii; blue in iv, viii) in P1 (from n = 4 biological replicates, both sexes) and P21 murine brains (n = 3 biological replicates, both sexes—see *Appendix A for images from additional biological replicates*). Scale bars are 250 microns. Note the diffuse signal associated with PDGFRβ immunostaining in addition to signals associated with vessels in ii, iv, vi, and viii. A select region of the P1 vessels is shown at higher magnification to visualize PDGFRβ signals associated with vessel walls and those in the parenchyma. Scale bar is 50 microns. (**B**) Representative confocal images of immunohistochemical labeling of PECAM-1 (i, green in v), PDGFRβ (ii, red in v), and GFAP (iii, cyan in v) with cell nuclei stained with DAPI (iv, white in v) in P1 murine brain (from n = 5 biological replicates, both sexes). Scale bars are 200 microns. Note the diffuse signal associated with PDGFRβ immunostaining in addition to signals associated with vessels in ii and iv.

**Figure 6 biomolecules-13-00711-f006:**
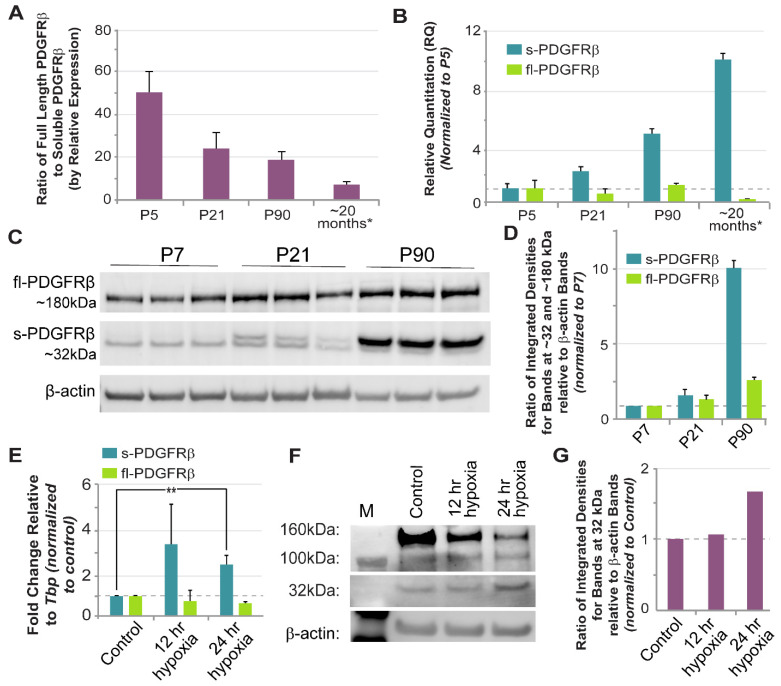
**Soluble PDGFRβ Transcript and Protein Levels Increased in the Murine Brain with Age and in a Cell-Based Model of Acute Hypoxia.** (**A**) Graph of the ratio of full-length *Pdgfrβ* transcripts relative to the soluble i10 isoform by relative expression using qRT-PCR to evaluate mRNA levels in murine brains at postnatal day 5 (P5), P21, P90, and at ~20 months. The housekeeping gene was *Tbp*. The asterisk (*) denotes samples taken from non-induced *Vhl* control littermates instead of WT, as described in the Methods section. Values are averages with error bars representing standard error of the mean. Biological replicates were n = 4–5 for all ages and represent both sexes. (**B**) Graph of the relative quantitation (RQ) of soluble i10 (teal) and full-length (green) *Pdgfrβ* in mouse brain at P5, P21, P90, and ~20 months, with values referenced to P5 for relative comparison across time (i.e., indicative of fold change). The asterisk (*) denotes samples taken from non-induced *Vhl* control littermates instead of WT, as described in the Methods section. Values are averages with error bars representing standard error of the mean. Biological replicates were n = 4–5 for all ages and represent both sexes. (**C**) Representative Western blot images of full-length PDGFRβ (fl-PDGFRβ, ~180 kDa) and soluble PDGFRβ (s-PDGFRβ, ~32 kDa) with β-actin shown for reference. Protein lysates were collected from murine brains at P7 (lanes 1–3, n = 3, both sexes), P21 (lanes 4–6, n = 3, both sexes), and P90 (lanes 7–9, n = 3, both sexes). (**D**) Graph of the ratios of integrated densities for 32 kDa (s-PDGFRβ) and 180 kDa (fl-PDGFRβ) bands relative to β-actin bands, with values referenced to P7 for relative comparison across time (i.e., indicative of fold change). (**E**) Relative quantitation of soluble i10 (teal) and full-length (green) *Pdgfrβ* referenced to control samples using qRT-PCR to evaluate mRNA levels from mouse embryonic stem cell (ESC)-derived vessels exposed to hypoxia 12 or 24 h before collection at day 10 differentiation (*Tbp* housekeeping gene). Values are averages with error bars representing standard deviation. n = 3–5 biological replicates for each condition. ** *p* ≤ 0.01 by ordinary one-way ANOVA followed by unpaired Tukey’s *t*-test (outlier data points removed by Grubbs’ test, or the extreme studentized deviate method, with α = 0.05). (**F**) Representative Western blot images of full-length PDGFRβ (fl-PDGFRβ, ~160 kDa) and soluble PDGFRβ (s-PDGFRβ, ~32 kDa) with β-actin (~50 kDa) shown for reference. Protein lysates were collected from differentiated ESCs under normoxia (control) or exposed to 12 or 24 h of hypoxia. n = 3–5 biological replicates for each condition. (**G**) Graph of the ratios of integrated densities for 32 kDa (s-PDGFRβ) bands relative to β-actin bands, with values referenced to Control (normoxia) for relative comparison across experimental groups.

## Data Availability

The data presented in this study are available on request from the corresponding author.

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
