# Peer review of "A Soluble Platelet-Derived Growth Factor Receptor-β Originates via Pre-mRNA Splicing in the Healthy Brain and Is Upregulated during Hypoxia and Aging"

_biomolecules, 2023, doi:10.3390/biom13040711_

Round 1

Reviewer 1 Report

In this manuscript, under normal, physiological settings, the authors discovered sPDGFR protein in the mouse brain and other tissues. Using brain samples for further investigation, the researchers discovered mRNA sequences corresponding to sPDGFR isoforms. Soluble PDGFR protein was observed in discrete locations of the brain parenchyma, such as the lateral ventricles, with signals being found more broadly near to cerebral microvessels, consistent with pericyte labeling. The authors also discovered that as mice aged, their transcript and protein levels increased, and that acute hypoxia raised sPDGFR variant transcripts in a cell-based model of intact vasculature. In general, these results are interesting, but there are several problems (see below) to be solved:

1.   Suggest improving the title “A Soluble Platelet-Derived Growth Factor Receptor- Originates via Pre-mRNA Splicing in the Healthy Brain and is Differentially Regulated during Hypoxia and Aging”,the article has confirmed“hypoxia and Aging can induce elevated production of sPDGFRβ isoform transcripts”.

2.   Suggest adding one keyword“Pre-mRNA Splicing”,because your article focuses on“sPDGFRβ via Pre-mRNA Splicing”.

3.   Introduction suggests to be brief and express your views clearly and concisely.

4.   Figure 1 Some of the GAPDH strips are not obvious and cannot be clearly judged.

5.   Some experimental results in the article are repeated less often, it is recommended to repeat more experiments, so as to make your article results more convincing.

6.   There is some disorder in the text layout, such as: “Aging and Acute Hypoxia Induce Increased Levels of sPDGFRβ Transcript and Protein”. Some of the pictures are not neatly arranged.

7.   Pay attention to how the abbreviation is written , such as“Co-Immunoprecipitation (Co-IP)”,Some in the article write "co-IP" and some write "Co-IP".

8.   It is recommended to further polish the grammar to read more fluently.

Reviewer 2 Report

This study explores the presence of a Soluble Platelet-Derived Growth Factor Receptor-b produced by Pre-mRNA alternative Splicing by pericytes of blood vessels in Various Murine Tissues with Enrichment in the Kidney and Brain. This soluble isoform retains PDGF-BB binding activity. This shortened transcript colocalizes with full-length receptor and with PECAM1 transcript. The IHC analyses also show Pdgfrβ-protein immunolabelling associated with blood vessels and perivascular ECM. The expression of soluble Pdgfrβ-transcript increases with the age of the mice and in a cell-based model of hypoxia where endothelial cells and pericytes differentiated from mouse embryonic stem cells were exposed to 3% oxygen. This study represents an improvement in comprehension of the pericyte role in vascular function and dysfunction.

Minor suggestions

Abstract: (RTK) is unnecessary; please modify the sentence in line 37-38

In Wb paragraph (143), please add the number and age of used mice for WB analyses;

In 147 Dubeco must be modified in Dulbecco.

Please add more information about human aorta smooth muscle cell lysates in Mat & Met.

THE PARAGRAPH TITLED ‘Pdgfrβ Splice Variant Transcripts Localize near Pecam1-positive Cells in the Murine Brain with PDGFRβ Protein Associated with Vessels and Interstitial ECM’ could be divided into two different paragraphs to enhance the readability

PECAM 1 IS THE PUTATIVE PERICYTE MARKER FOR THE UNIQUE SUBSET OF MIDBRAIN PERICYTYES ORIGINATING FROM A MONOCYTE/MACROPHAGE POPULATION OF THE EMBRYONIC YOLK SAC

https://www.nature.com/articles/s41598-017-03994-1

please modify the sentence in 437

FIG 4 IS QUITE INSUFFICIENT FOR DEMONSTRATING TRIPLE COLOCALIZATION OF THE SIGNALS, PLEASE INCREASE THE MAGNIFICATION IN INSETS OR ADDITIONAL FIGURES. In addition, the anatomical location of figure 4v-viii is CHOROIDAL PLEXUS along the floor of the third ventricle. Please add figures of blood vessel of the cerebral cortex and white matter.

Distinct RED or distinct GREEN signals in 4iii e 4vii are not visible in my pdf copy; please increase the magnification and choose other images to show also distinct vascular profiles with elongated shape of nuclei. If possible, please, SHOW ALSO sPdgfrβ-associated signals differentiated from fl-Pdgfrβ signals.

Please quantify the colocalizing signals of sPdgfrβ with fl-Pdgfrβ and with PECAM1.

In Fig 5 there are two similar images of the same region of the brain, triple immunolabelling could be sufficient, but I suggest to show also a high power inset showing the wall of a few vessels.

Why have you used P21 mice for WB, RNAscope and P1 mice for IHC? In Mat &Met it is affirmed the same age. Please show an image from P21 brain or explain the choice to analyze only P1.

Fig 6 mos for months is unusual; please modify.

Please add the age of the mice used for demonstrating the presence of soluble PDGFRbeta at line333

In Fig 6, Please add a Graph of the ratio of full-length Pdgfrβ/tubulin and a Graph of the ratio of 32kDa sPdgfrβ/tubulin.

In Fig 6, Please add a Representative Western Blot and a representative microscopic image of the experiment with mouse embryonic cell-derived vessels in one of the hypoxic conditions.

At this magnification, it is impossible to localize Pdgfrβ protein by IHC in astrocyte, please modify the sentence in 472 and 627, because it is a speculation, in my opinion.

Reviewer 3 Report

In this study, the authors show sPDGFRβ originates from pre-mRNA splicing. It is an important study to understand its specific role in other neurodegenerative diseases, which may also help to study possible treatment strategies.

A few points need to be addressed,

Did the authors try exposing hypoxia on mice and then found out how the sPDGFRβ change in the tissue?

Since the authors study mainly the sPDGFRβ, they need to include more studies in the introduction and discussion which may have published in other neurodegenerative disease conditions if any, to highlight its importance, also give supporting references.

In this study the authors did not maintain consistency of using one particular gender or both in the experiments. For example, in western blot and qRT-PCR experiments they mentioned both male and female mice, in RLM-RACE females were used, in Co-PI male mice were used, and did not mention what mice were used both in Fluorescent RNA-scope and immunohistochemistry?

In those experiments were the authors used both the genders, did they see any differences between the genders?

Also, In the results authors mentioned about the number of animals they have used for each experiments, some of the results they have n=1, it is important to report the results with atleast n=3 minimum for each experiments.

Round 2

Reviewer 3 Report

Dear Authors,

The authors tried to address some of the concerns in the revised version of their manuscript. Eventhough, the authors have their reason for why they do not have enough number of animals in some of their experiments, it is not appropriate to come with a conclusion in the results from just one animal number and there for suggesting them to do enough experiments with atleast to reach minimum required sample size.
